# Assessment of Delirium Using the Confusion Assessment Method in Older Adult Inpatients in Malaysia

**DOI:** 10.3390/geriatrics4030052

**Published:** 2019-09-11

**Authors:** Hui Min Khor, Hwee Chin Ong, Bee Kuan Tan, Chung Min Low, Nor’Izzati Saedon, Kit Mun Tan, Ai Vyrn Chin, Shahrul B. Kamaruzzaman, Maw Pin Tan

**Affiliations:** 1Department of Medicine, Faculty of Medicine, University of Malaya, Kuala Lumpur 59100, Malaysia; izzati@ummc.edu.my (N.S.); kmtan@ummc.edu.my (K.M.T.); avchin@ummc.edu.my (A.V.C.); mptan@ummc.edu.my (M.P.T.); 2Taylor’s University, Faculty of Health and Medical Sciences, Selangor 47500, Malaysia; hweechin33@hotmail.com (H.C.O.); beekuan328@gmail.com (B.K.T.); 3Department of Medicine, Hospital Sultan Ismail, Johor, Johor Bahru 81100, Malaysia; lowchungmin@gmail.com; 4Department of Medical Sciences, School of Healthcare and Medical Sciences, Sunway University, Bandar Sunway, Selangor 47500, Malaysia

**Keywords:** delirium, older, confusion, inpatient, hospital, Malaysia

## Abstract

The detection of delirium in acutely ill older patients is challenging with the lack of informants and the necessity to identify subtle and fluctuating signs. We conducted a cross-sectional study among older patients admitted to a university hospital in Malaysia to determine the presence, characteristics, and mortality outcomes of delirium. Consecutive patients aged ≥65years admitted to acute medical wards were recruited from August to September 2016. Cognitive screening was performed using the mini-mental test examination (MMSE) and the Confusion Assessment Method (CAM). The CAM-Severity (CAM-S) score was also performed in all patients. Of 161 patients recruited, 43 (26.7%) had delirium. At least one feature of delirium from the CAM-S short and long severity scores were present in 48.4% and 67.1%, respectively. Older age (OR: 1.07, 95% CI: 1.01–1.14), immobility (OR: 3.16, 95% CI: 1.18–8.50), cognitive impairment (OR: 5.04, 95% CI: 2.07–12.24), and malnutrition (OR: 3.37; 95% CI: 1.15–9.85) were significantly associated with delirium. Older patients with delirium had a higher risk of mortality (OR: 7.87, 95% CI: 2.42–25.57). Delirium is common among older patients in our setting. A large proportion of patients had altered mental status on admission to hospital although they did not fulfill the CAM criteria of delirium. This should prompt further studies on strategies to identify delirium and the use of newer, more appropriate assessment tools in this group of vulnerable individuals.

## 1. Introduction

Delirium is a complex clinical condition characterized by acute onset of disturbance in consciousness, cognitive fluctuation, inattention, and perception [1,2]. Delirium is common among hospitalized older people [3,4]. The prevalence of delirium among older adults upon admission to hospital is 10–30% with a further 3–29% arising as new cases during their hospital stay [5]. Delirium in older adults is associated with serious adverse outcomes including prolonged hospitalization, increased healthcare cost, institutionalization, functional and cognitive decline, and increased mortality risk [6,7,8].

Despite its serious consequences, delirium is frequently undetected and misdiagnosed [9,10,11]. Delirium has a multifactorial aetiology and a fluctuating course. The diagnosis of delirium can only be made using clinical skills with careful history-taking from family members or carers to identify the presence of any acute changes in cognitive function [12]. Risk factors that predispose the older adult to the development of delirium include advanced age, male gender, pre-existing dementia and depression, visual and hearing impairment, functional dependence, dehydration and malnutrition, polypharmacy, alcohol excess, and coexistence of multiple and severe medical conditions [13,14,15]. Early identification of these vulnerable patients with the above risk factors may allow intervention to be held in place to prevent the development of delirium during hospital admissions. A handful of screening tools has been validated for the identification of delirium in different healthcare setting such as in intensive care units, geriatric units, palliative care, surgical units, emergency departments, and residential care settings [16,17,18]. An ideal screening tool for delirium should be brief, require little or no training, and be appropriate to the clinical setting it is used in.

A recent study suggested that 25% of the total attendances to an emergency department in a teaching hospital in Kuala Lumpur are aged 60 years and older [19]. Older individuals subsequently make up fifty percent of all admissions to the hospital ward. The burden of delirium among older patients in Malaysia has not previously been evaluated, with no previous efforts in assessing the effectiveness of existing delirium assessment tools in a multi-ethnic country. We therefore identified the presence of delirium, factors associated with delirium, and mortality outcomes in individuals with delirium in a teaching hospital in Malaysia.

## 2. Materials and Methods

This was a cross sectional study involving consecutive older patients aged 65 years and above admitted to the medical wards of the University of Malaya Medical Centre, Kuala Lumpur from August to September 2016. All patients included in the study were assessed within 48 h of admission. Individuals who were severely ill requiring ventilatory or inotropic support and individuals with severe hearing impairment deemed unsuitable to be assessed by the medical team were excluded from the study. This study was performed as part of a quality assurance project and did not require ethical approval as advised by the Medical Ethics Committee of the University of Malaya Medical Centre.

The patients were recruited by trained researchers, comprised of medical undergraduates and graduates during their research attachment and verified by consultant geriatricians. All researchers had received a one-day training on delirium and the assessment of delirium. They would have conducted at least one assessment under direct supervision, and were only allowed to conduct assessments independently once considered competent by the supervisor.

Details on basic socio-demographic, place of residence, education level, and pre-existing comorbidities were obtained from patient records. The predisposing factors for delirium such as presence of prior cognitive impairment, visual or hearing impairment and immobility, as well as precipitating factors which include dehydration, hypoxia, infection, pain, and sleep disturbance were also recorded [3,20]. The presence of prior cognitive impairment was determined through previous formal medical diagnosis of dementia of various types, prior documentation of cognitive impairment (unspecified subtype) in the medical records or history of pre-existing, uninvestigated cognitive impairment for at least a one year period reported by relatives or caregivers. Nutritional status was assessed using the Mini Nutritional Assessment score [21]. Hospitalization outcomes including length of stay and inpatient mortality were obtained from the hospital electronic records.

### 2.1. Cognitive Screening Test

Cognitive screening tests were performed for all patients using the mini mental test examination (MMSE). The MMSE is a 30-point assessment scale widely used in the diagnosis and monitoring of cognitive function in the elderly patients. It examines the cognitive domains of registration, attention and calculation, recall, language, ability to follow simple commands, and orientation. Individuals who were unable to perform the MMSE due to absence of any meaningful interactions were given a score of 0. The presence of cognitive impairment was defined as an MMSE score of 25 or lower.

### 2.2. Confusion Assessment Method

The Confusion Assessment Method (CAM) was performed for all patients with or without suspected delirium. CAM has a sensitivity of 94–100%, and specificity of 90–95% in diagnosing delirium in older persons [2]. It includes a diagnostic algorithm, based on four cardinal features of delirium: (1) acute onset and fluctuating course; (2) inattention; (3) disorganized thinking; and (4) altered level of consciousness. The diagnosis of delirium by CAM requires the presence of both the first and second feature and at least one of the other two. The severity of delirium was evaluated by employing the CAM-S short and long form. To score the CAM-S short form, the 4 core features of the Confusion Assessment Method (CAM) were rated with a severity score which yields a sum ranging from 0 to 7 (7 = most severe). CAM-S long form has additional features which includes disorientation, memory impairment, perceptual disturbances, psychomotor agitation, psychomotor retardation and altered sleep–wake cycle. Scores for CAM-S long range from 0–19 where higher scores indicating more severe delirium.

Each item of the CAM and CAM-S short and long scales were assigned a score by the trained assessor or if the patient is unable to provide any meaningful response, the item is marked ‘untestable’. Each untestable item is then given a full score for subsequent analysis. Therefore, if an individual scored had evidence of acute onset fluctuating course and altered level of consciousness, but could not be assessed for inattention and disorganized thinking due to the presence of stupor and coma, the assumption is made that they had severe inattention and disorganized thinking. The individual would therefore be included in the delirium group within the CAM algorithm classification. In addition, within the CAM-S short and long form scoring, the individual will be assigned the maximum score of 2 for the item for which assessment was not possible due to absence of meaningful communication. Individuals with a total CAM-S short form score of 0 were considered unlikely to have delirium, those with scores of 1 to 2 were considered possible delirium, and 3–7 probable delirium. Individuals with a total CAM-S long form score of 0 were considered unlikely to have delirium, those with scores of 1–4 possible delirium and 5–19 probable delirium [22].

### 2.3. Statistical Analysis

Data collection and analyses were performed using the Statistical Package for Social Sciences (SPSS, IBM Ltd. USA) version 23. Age-standardized prevalence of delirium was estimated according to the age-characteristics of the overall medical inpatient population over the study period. The normally distributed continuous variables were presented as mean (standard deviation), while categorical variables were presented as frequency and proportion. Descriptive analyses were performed on demographic characteristics, comorbidities, and risk factors for delirium. The categorical variables were compared between groups using chi-square test. Comparisons were made between individuals in the delirium and non-delirium group. A separate analysis was performed comparing individuals who were untestable with those in whom it was possible to perform a structured CAM assessment. Between-group comparisons were performed using the independent t-test and Mann-Whitney U test for normally and non-normally distributed data, respectively. The correlation between the CAM-S short form and long form with the MMSE was calculated using Spearman’s correlation. Any variables associated with delirium that yielded a p-value of less than 0.10 from the initial analysis were entered into the regression model for mortality outcome. Statistical significance was set at p-value less than 0.05.

## 3. Results

### 3.1. Prevalence of Delirium

A total of 398 individuals aged 65 years and over were admitted to the medical wards of our institution within the two-month period. A total of 178 older in-patients were assessed within 48 h of admission. The remainder was not included due to difficulties in accessing the patient within the 48 h of admission due to patient movement and limitations in staffing. There were no age and gender differences between those assessed and not assessed (Table A1). Seventeen patients were excluded from the study due to various reasons: refusal to participate (n = 6), terminally ill (n = 7), severe hearing impairment (n = 1), and missing data (n = 3) as shown in Figure 1. Hence, the final sample comprised of 161 patients. The characteristics of the patients included in the study, with and without delirium, are shown in Table 1.

Of the 161 individuals assessed, 30 (18.6%) had fulfilled the criteria for delirium according to the CAM screening tool algorithm. An additional 13 patients (8.1%) had acute onset of altered level of consciousness according to their next of kin during the interview, but the assessors were unable to illicit any meaningful communication to complete either the CAM or MMSE assessment. These 13 individuals were also included as individuals with delirium within the CAM algorithm. In the CAM-S short and long forms these individuals were assigned the maximum score of 2 in items in which assessment was not possible due to stupor or coma (Table 2). Hence a total of 43/161 (26.7%) patients were suspected to have delirium in our population setting. Age-standardized adjustment revealed an estimated prevalence of delirium of 25.4% using the CAM algorithm in all medical inpatients aged 65 years and above. In addition, 78 (48.4%) had possible (CAM Score 1–2) or probable delirium (CAM Score 3–7) according to the CAM-S short form assessment, while 108 (67.1%) had possible (CAM Score 1–4) or probable (CAM Score 5–19) delirium according to the CAM-S long form assessment. The age-standardized prevalence for possible or probable delirium according to CAM-S short and long form assessments were 46.9% and 65.5% respectively.

### 3.2. Association between MMSE and CAM

Table 3 summarizes the mean MMSE scores for individuals for whom delirium was unlikely, or for those with possible or probable delirium for the CAM-S short and long form assessments. The MMSE score was significantly correlated with both the CAM-S short and long form assessment scores.

### 3.3. Risk Factors for Delirium

In the adjusted univariate analysis from Table 1, patients with delirium were of an older age group (81.5 (SD 8.6) vs. 74.7 (SD 6.6), *p* < 0.01), less likely to have chronic obstructive pulmonary disease (OR 0.11, 95% CI: 0.01–0.87) and more likely to have prior cognitive impairment (OR 5.78, 95% CI: 2.53–13.22), immobility (OR 3.66, 95% CI: 1.54–8.65), and malnutrition (OR 5.53, 95% CI: 2.05–14.93). In the multivariable regression analysis, all the factors that yielded a p value of <0.10 were entered into the model. Of these, older age, cognitive impairment, immobility, and malnutrition were independently associated with delirium (Table 4).

### 3.4. Hospitalization Outcomes

The average length of hospital stay was significantly higher in patients with delirium (13.1 (SD 12.6) days vs. 7.5 (SD 8.2) days, *p* < 0.01). Nineteen patients (11.8%) died in hospital. Patients with delirium were also at higher risk of in-patient mortality (age adjusted odds ratio = 7.87; 95% CI 2.42–25.57; *p* < 0.01).

## 4. Discussion

The present study showed that delirium was present in 26.7% of older patients on admission to medical wards at our middle-income developing country. This finding was comparable to studies performed in medical wards of our neighbouring Asian countries of Singapore and India, with results of 28.1% and 27.5% respectively [23,24]. A much higher percentage was found in a university hospital in Thailand where 40.4% of older medical inpatients had delirium on admission [25]. From all the three studies, a large proportion of patients with delirium were detected on admission and a smaller percentage (14.5% in India, 12.3% in Singapore and 8.4% in Thailand) developed it after hospitalization. This highlights the importance of prompt identification of at risk patients as soon as they are admitted for proper management and implementation of effective delirium preventative strategies.

As there is limited published data on the characteristics of these at risk older patients admitted to the medical wards in Malaysia, we investigated the risk factors and clinical outcomes of older patients with delirium admitted to the medical wards in a large university hospital in Malaysia using available delirium and cognitive assessment tools. Despite the widespread use of the CAM as screening tool for delirium in hospital, the tools’ scoring mechanism does not allow categorization of patients who presents to acute hospitals with reduced consciousness (not comatose) that affect their ability to engage with cognitive testing or interview [26]. This group of “untestable” patients with drowsiness, stupor, obtundation, or agitation has multiple negative outcomes and mortality risk [27,28]. The European Delirium Association and American Delirium Society suggest that it is clinically safer to include these patients within the delirium group until proven otherwise [29].

Cognitive assessment in the elderly is often overlooked during acute medical admissions as healthcare professionals have a tendency to focus on physical rather than mental health [10,30,31]. With the rising proportion of the aged population in Malaysia, the number of dementia and cognitive impairment cases is expected to increase [32]. Advanced age, cognitive impairment, immobility and malnutrition were associated with delirium in our study. These are well known risk factors for delirium as shown in the literature. Although delirium has been defined as an acute and frequently reversible condition, emerging evidence suggests that it could be associated with incomplete recovery [33,34]. Further deterioration in cognitive and physical function following delirium among older persons post discharge has been reported [35,36].

In our study, 48.4% and 67.1% of patients had at least one feature of delirium on presentation to hospital from the CAM-S short and long severity scores respectively. The severity has also been shown to be inversely related to MMSE scores. This highlights that the patients had at least one feature of delirium on presentation to hospital which may be due to underlying dementia or a sub-syndromal presentation of delirium. Sub-syndromal delirium has been reported as an intermediate stage between delirium and normal mental states which may or may not progress to full delirium [37,38,39]. There is currently no consensus definition of the condition. The outcome of patients with sub-syndromal delirium is associated with poorer prognosis, similar to those of delirium [40,41,42,43]. A systematic review of elderly patients with sub-syndromal delirium found that older age, dementia, and basic activities of daily living impairment, more severe physical illness, and more comorbidities were risk factors of the condition [44]. Irrespective of the different diagnosis of cognitive impairment (delirium alone, dementia, delirium superimposed on dementia or unspecified cognitive impairment), Reynish et al. [45] found that hospitalized older patients with cognitive impairment had worse outcomes such as longer hospitalization, readmissions, and mortality compared to those without cognitive impairment.

From the present study, a third of the patients with suspected delirium had altered level of consciousness that affected their ability to engage with cognitive testing during our assessment. Altered mental status is a term to describe disorder of mental functioning which could range from slight confusion to coma [46]. Acute changes in mental status may represent acute brain dysfunction secondary to delirium, stupor, and coma. There is significant overlap between these spectrums and transition between delirium-to-coma is possibly a continuum of progressively lesser degrees of arousal [47]. Many studies on delirium have excluded this category of patients which may lead to the underestimation of delirium rates in severely ill elderly patients [26,48,49]. In a recent study, Aslander et al. [50] investigated the aetiology of elderly patients presenting to emergency department with altered mental status and the coexistence of delirium in these patients. They found that delirium was highest in coma-stupor patients at 75.9%, followed by 60% in awake-hyperalert patients, and 52.0% in lethargic patients. The rate of ICU admission and mortality was highest in the coma-stupor group of patients. The use of delirium screening tool such as the 4AT rapid assessment test which allows the assessment of “untestable” patients may help to increase diagnostic accuracy for this group of patients [51,52,53].

This study identifies that delirium is common and is associated with advanced age, cognitive impairment, immobility, and malnutrition. Despite concerted efforts to conduct consecutive sampling, the rapid turnover within the acute medical wards of our hospital had led to an identification rate of below 50%. We were, however, able to obtain an accurate list of all older patients admitted during the study period, alongside their basic demographics from hospital records. It was therefore possible to estimate the overall prevalence of delirium by calculating the adjustment weights of the base population over the actual population. The assessment of delirium was performed using the CAM alone which may have led to under reporting of delirium cases. The use of other delirium screening tools such as the newer 4AT would help to include the “untestable” patients whose arousal is too abnormal to have attention assessed by interview. Lastly, this study was performed during the first 48 h of admission and patients were only assessed once during which the data was collected. As delirium is a fluctuating condition, some cases of delirium would have been missed.

## 5. Conclusions

One in four older patients admitted to the acute medical wards in a university hospital in Malaysia has delirium. The presence of delirium was associated with individuals of advanced age, poor functional status on admission, prior history of cognitive impairment and malnutrition. Early identification of at risk individuals is important to allow for the implementation of multifaceted interventions to prevent delirium in this vulnerable group of patients. Further studies should be performed to identify more suitable delirium assessment tools for our specific population, as well as to determine the value of cognitive screening among older hospitalized individuals.

## Figures and Tables

**Figure 1 geriatrics-04-00052-f001:**
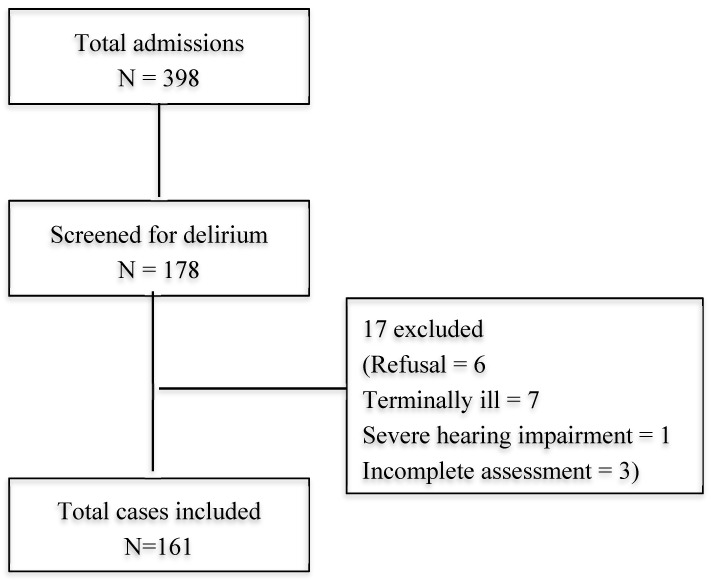
Flow Diagram of Study Recruitment.

**Table 1 geriatrics-04-00052-t001:** Characteristics of patients with and without delirium.

Variables	Total (n = 161)	Delirium (n = 43)	No Delirium (n = 118)	Unadjusted OR (95% CI)	Adjusted OR ^†^ (95% CI)
Age (mean, SD)	76.5 (7.77)	81.53 (8.62)	74.66 (6.57)		
Male Gender n (%)	83 (51.6)	22 (51.2)	61 (51.7)	0.98 (0.49–1.97)	1.23 (0.57–2.66)
Ethnicity					
Chinese	82 (50.9)	26 (60.5)	56 (47.5)	0.63 (0.24–1.64)	0.78 (0.28–2.22)
Malay	42 (26.1)	9 (20.9)	33 (28)	0.59 (0.25–1.40)	0.84 (0.33–2.18)
Indian	31 (19.3)	7 (16.3)	24 (20.3)	0.43 (0.05–3.88)	0.14 (0.01–1.81)
Nursing Home	10 (6.2)	5 (11.6)	5 (4.2)	2.97 (0.82–10.83)	1.50 (0.35–6.38)
<6 years Education	103 (64)	25 (59.5)	78(66.1)	0.75 (0.37–1.56)	0.99 (0.45–2.20)
**Comorbidities**					
CCF	13 (8.1)	5 (11.6)	8 (6.8)	1.81 (0.56–5.87)	1.93 (0.54–6.88)
CKD	36 (22.5)	10 (23.3)	26 (22.2)	1.06 (0.46–2.44)	1.23 (0.48–3.12)
Dementia	16 (9.9)	7 (16.3)	9 (7.6)	2.36 (0.82–6.78)	1.57 (0.49–4.98)
Diabetes Mellitus	74 (46.3)	20 (46.5)	54 (46.2)	1.01 (0.50–2.05)	1.17 (0.54–2.51)
IHD	46 (28.6)	9 (20.9)	37 (31.4)	0.58 (0.25–1.33)	0.61 (0.25–1.50)
Hypertension	110 (68.8)	28 (65.1)	82 (70.1)	0.80 (0.38–1.67)	0.77 (0.34–1.72)
COPD	20 (12.5)	1 (2.3)	19 (16.2)	0.12 (0.02–0.95)	0.11 (0.01–0.87)
Osteoarthritis	15 (9.3)	1 (2.3)	14 (11.9)	0.18 (0.02–1.39)	0.16 (0.02–1.28)
Stroke	33 (20.5)	12 (27.9)	21 (17.8)	1.79 (0.79–4.05)	1.95 (0.8–4.74)
**Risk Factors**					
Cognitive impairment	45 (28)	26 (60.5)	19 (16.1)	7.97 (3.64–17.45)	5.78 (2.53–13.22)
Dehydration	72 (44.7)	26 (60.5)	46 (39)	2.39 (1.17–4.89)	1.69 (0.77–3.68)
Hypoxia	67 (41.6)	22 (51.2)	45 (38.1)	1.70 (0.84–3.44)	1.57 (0.73–3.38)
Infection	89 (55.3)	29 (67.4)	60 (50.8)	2.00 (0.96–4.17)	1.42 (0.64–3.14)
Immobility	84 (52.2)	34 (79.1)	50 (42.4)	5.14 (2.26–11.67)	3.66 (1.54–8.65)
Malnutrition (MNA < 8)	83 (51.9)	36 (85.7)	47 (39.8)	9.06 (3.54–23.19)	5.53 (2.05–14.93)
Pain	73 (45.3)	18 (41.9)	55 (46.6)	0.82 (0.41–1.67)	1.09 (0.51–2.35)
Sensory impairment	27 (16.8)	9 (20.9)	18 (15.3)	1.47 (0.60–3.58)	1.02 (0.38–2.74)
Sleep disturbance	69 (42.9)	19 (44.2)	50 (42.4)	1.08 (0.53–2.18)	1.31 (0.61–2.82)

SD = standard deviation, OR = odds ratio, CCF = congestive cardiac failure, CKD = chronic kidney disease, IHD = ischemic heart disease, MNA = mini nutritional assessment. ^†^ adjusted for age with logistic regression.

**Table 2 geriatrics-04-00052-t002:** Summary of Individual Item Scores and Percentages.

CAM	Severity Score n (%)			Unable to Assess n (%) ^‡^
	0	1	2	
Item 1				
Acute onset and fluctuating course	109 (67.7)	52 (32.3)	n/a	0
Item 2				
Inattention	101 (62.7)	39 (24.2)	21 (13)	13 (8.1)
Item 3				
Disorganised thinking	118 (73.3)	17 (10.6)	26 (16.1)	13 (8.1)
Item 4				
Altered level of consciousness	110 (68.3)	38 (23.6)	13 (8.1)	0
Item 5				
Disorientation	97 (60.2)	33 (20.5)	31 (19.3)	13 (8.1)
Item 6				
Memory impairment	84 (52.2)	43 (26.7)	33 (20.5)	13 (8.1)
Item 7				
Perceptual Disturbances	142 (88.2)	12 (7.5)	7 (4.3)	2 (1.2)
Item 8				
Psychomotor agitation	135 (83.9)	22 (13.7)	4 (2.5)	0
Item 9				
Psychomotor retardation	137 (85.1)	13 (8.1)	11 (6.8)	0
Item 10				
Altered sleep-wake cycle	112 (69.6)	30 (18.6)	19 (11.8)	0

^‡^ Individuals in this category were subsequently assigned a maximum score of 2.

**Table 3 geriatrics-04-00052-t003:** Confusion Assessment Method-Severity (CAM-S) Short and Long Form and mini-mental test examination (MMSE) scores.

Assessment Tools	Delirium Unlikely	Possible Delirium	Probable Delirium	Spearman’s Rho	*p*-Value
**CAM-S Short**					
Total, n (%)	83 (51.6)	30 (18.6)	48 (29.8)	−0.77	<0.01
MMSE (mean, SD)	23.19 (6.32)	14.27 (8.83)	3.65 (5.12)		
**CAM-S Long**					
Total, n (%)	53 (33.9)	53 (32.9)	55 (34.2)	−0.80	<0.01
MMSE (mean, SD)	25.32 (5.10)	17.98 (7.50)	4.24 (5.74)		

**Table 4 geriatrics-04-00052-t004:** Multivariable analysis on factors associated with delirium in the elderly.

Characteristics	B coefficient	SD	OR (95% CI)
Age	0.07	0.03	1.07 (1.01–1.14)
Immobility	1.15	0.50	3.16 (1.18–8.50)
Malnutrition	1.22	0.55	3.37 (1.15–9.85)
Cognitive impairment	1.62	0.45	5.04 (2.07–12.24)

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
