# Peer review of "Assessment of Delirium Using the Confusion Assessment Method in Older Adult Inpatients in Malaysia"

_geriatrics, 2019, doi:10.3390/geriatrics4030052_

Round 1

Reviewer 1 Report

In the paper "Assessment of delirium using the Confusion Assessment Method in older adult inpatients in Malaysia" the authors present an observational study reporting the prevalence of delirium in older patients admitted to acute medical wards. The manuscript is easy to follow and well structured. It addresses an important topic and the data obtained is relevant to this field.

I have only one minor comment: in the abstract the authors state correctly that this is a cross-sectional study. However, in the Methods section it is argued that the study is prospective, which is not the case (patients were assessed only one time).

Author Response

Response to Reviewer 1 Comments

In the paper "Assessment of delirium using the Confusion Assessment Method in older adult inpatients in Malaysia" the authors present an observational study reporting the prevalence of delirium in older patients admitted to acute medical wards. The manuscript is easy to follow and well structured. It addresses an important topic and the data obtained is relevant to this field.

Point 1: I have only one minor comment: in the abstract the authors state correctly that this is a cross-sectional study. However, in the Methods section it is argued that the study is prospective, which is not the case (patients were assessed only one time).

Response 1: Apologies for the mistake. This has now been rectified in the manuscript. (Page 3, Line 69).

Reviewer 2 Report

Thank you for the opportunity to review this paper on delirium in older Malaysian inpatients. I have a number of comments for the authors to consider, divided into major and minor.

MAJOR

The authors correctly identify the study as cross-sectional in the abstract, however in the paper it is referred to as a "prospective observational study". There was no follow-up of patients, therefore this is a cross sectional study. To this end, use of the terms "risk factor" and "predictor" should be removed and the term association used instead. Both the introduction and discussion are sparsely cited. There are numerous statements that require referencing (e.g., the first two sentences of the paper). Please carefully reference these sections.

MINOR

The aim of the paper is not clearly articulated. It is unclear what the authors mean by the term "utility". Could the authors please clarify the statement on Page 2, Lines 60-63, that the burden of delirium has not been assessed in "this setting". What setting are they referring to? The need for this study needs to be more clearly stated. More information on recruitment is required in the methods. How were individuals designated "severely ill" (Page 3, Line 71)? How was the presence of prior cognitive impairment determined (Page 3, Line 77)? Please describe what the groupings for the variables listed on Page 3, Lines 76-79 were determined. Please describe the qualifications and training of the delirium assessors. Please provide a participant flow diagram that describes reasons for exclusion. In Table 4, please clarify that the estimates in the 4th column are odds ratios. Providing both confidence intervals and p-values is redundant. In places where both are presented the confidence intervals would be sufficient. The sentence on Page 9, Lines 198-199 is unclear. What is the comparison group? What are the units? The term "multivariate" should be replaced with "multivariable", given the the analyses conducted. In the discussion, Page 12, Lines 290-281, how did this study highlkight that cognitive assessment was not routinely performed? Please remove the word "mild" from Line 293; given the rationale presented, any case of delirium may have been missed. The first time frailty and adverse events are mentioned is in the conclusion. Please discuss earlier or remove.

Author Response

Thank you for the opportunity to review this paper on delirium in older Malaysian inpatients. I have a number of comments for the authors to consider, divided into major and minor.

MAJOR

Point 1: The authors correctly identify the study as cross-sectional in the abstract, however in the paper it is referred to as a "prospective observational study". There was no follow-up of patients, therefore this is a cross sectional study.

Response 1: Apologies for the mistake. This has now been rectified in the manuscript. (Page 3, Line 69).

Point 2: To this end, use of the terms "risk factor" and "predictor" should be removed and the term association used instead.

Response 2: Many thanks for the advice. The terms “risk factors” and “predictor” have been replaced with association instead. (Page 1,Line 28) (Page 10, Line 225) (Page 11, Line 264).

Point 3: Both the introduction and discussion are sparsely cited. There are numerous statements that require referencing (e.g., the first two sentences of the paper). Please carefully reference these sections.

Response 3: We apologize for the lack of referencing. We have revised and updated the referencing in our introduction and discussion accordingly.

MINOR

Point 4: The aim of the paper is not clearly articulated. It is unclear what the authors mean by the term "utility". Could the authors please clarify the statement on Page 2, Lines 60-63, that the burden of delirium has not been assessed in "this setting". What setting are they referring to? The need for this study needs to be more clearly stated.

Response 4: Thank you for the comments. We have removed this statement to avoid contention and the wording used for the aim of the study has been refined as below.

“A recent study suggested that 25% of the total attendances to an emergency department in a teaching hospital in Kuala Lumpur are aged 60 years and older [19]. Older individuals subsequently make up fifty percent of all admissions to the hospital ward. The burden of delirium among older patients in Malaysia has not previously been evaluated, with no previous efforts in assessing the effectiveness of existing delirium assessment tools in a multi-ethnic country. We therefore identified the presence of delirium, factors associated with delirium and mortality outcomes in individuals with delirium in a teaching hospital in Malaysia. “

(Page 2, Para. 3, Line 59-66).

Point 5: More information on recruitment is required in the methods. How were individuals designated "severely ill" (Page 3, Line 71)?

Response 5: Individuals who were “severely ill” include those who were critically ill requiring ventilatory and inotropic support. (Page 3, Line 72) A participant flow diagram has been included in the manuscript for clarification. (Page 6, Line 165).

Point 6: How was the presence of prior cognitive impairment determined (Page 3, Line 77)?

Response 6: The presence of prior cognitive impairment was determined through either prior documentation of cognitive impairment (unspecified subtype) in the medical records or history of pre-existing underlying cognitive impairment for at least one-year period reported by relatives or caregivers. The awareness of dementia in Malaysia is poorer than in other developed settings. However even in developing nations, undiagnosed dementia is still common. Therefore, we are unable to base the determination of prior cognitive impairment on clinical diagnosis. We apologize for this lack of clarity and have revised the relevant section of the manuscript accordingly. (Page 3, Line 87-91).

Point 7: Please describe what the groupings for the variables listed on Page 3, Lines 76-79 were determined.

Response 7: The variables listed have been grouped as predisposing and precipitating factors for delirium. (Page 3, Line 84).

Point 8: Please describe the qualifications and training of the delirium assessors.

Response 8: The patients were recruited by trained researchers which comprise of medical undergraduates and graduates during their research attachment and verified by consultant geriatricians. All researchers had received attended a one-day training on delirium and the assessment of delirium. They would have conducted at least one assessment under direct supervision, and only allowed to conduct assessments independently once considered competent by the supervisor. (Page 3, Line 77-82).

Point 9: Please provide a participant flow diagram that describes reasons for exclusion.

Response 9: We are grateful to this reviewer for their suggestion. A study flow chart has been included in the manuscript. (Figure 1).

Point 10: In Table 4, please clarify that the estimates in the 4th column are odds ratios. Providing both confidence intervals and p-values is redundant. In places where both are presented the confidence intervals would be sufficient.

Response 10: Many thanks for the comment. The p-values in Table 4 have been removed.

Point 11: The sentence on Page 9, Lines 198-199 is unclear. What is the comparison group? What are the units?

Response 11: From the adjusted univariate analysis, individuals with delirium were compared with individuals without delirium. The units have been included. (Page 10, Line 219).

Point 12: The term "multivariate" should be replaced with "multivariable", given the the analyses conducted.

Response 12: Thank you for the comment. This has now been corrected. (Page 10, Line 223).

Point 13: In the discussion, Page 12, Lines 290-281, how did this study highlkight that cognitive assessment was not routinely performed?

Response 13: Apologies for the confusion caused. The statement has been removed from the discussion. The study was also performed as an audit to assess the percentage of cognitive assessment performed by medical officers for elderly patients admitted to the hospital. Cognitive assessment through history taking from patients’ informant or the use of cognitive screening tools such as abbreviated mental test 4 (AMT4) or MMSE was identified as routine cognitive assessment performed. As the manuscript was not aimed at highlighting the results of the audit, the initial statement has been removed to avoid confusion to the readers.

Point 14: Please remove the word "mild" from Line 293; given the rationale presented, any case of delirium may have been missed.

Response 14: Many thanks for the comment. The word “mild” has been removed. (Page 13, Line 314).

Point 15: The first time frailty and adverse events are mentioned is in the conclusion. Please discuss earlier or remove.

Response 15: Once again, thank you for the comments. The terms “frailty” and “adverse events” have been removed.